# The Issues with Journal Issues: Let Journals Be Digital Libraries

C. Sean Burns 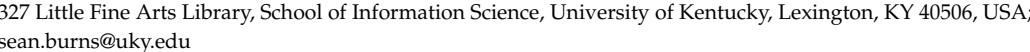

327 Little Fine Arts Library, School of Information Science, University of Kentucky, Lexington, KY 40506, USA; sean.burns@uky.edu

**Abstract:** Science depends on a communication system, and today, that is largely provided by digital technologies such as the internet and web. Despite the fact that digital technologies provide the infrastructure for this communication system, peer-reviewed journals continue to mimic workflows and processes from the print era. This paper focuses on one artifact from the print era, the journal issue, and describes how this artifact has been detrimental to the communication of science, and therefore, to science itself. To replace the journal issue, this paper argues that scholarly publishing and journals could more fully embrace digital technologies by creating digital libraries to present and organize scholarly output.

**Keywords:** journals; digital publishing; scholarly publishing; open science

## 1. Introduction

Science depends on a communication system. The internet and the web provide the infrastructure and tools to support that communicative system. Yet, despite the fact that they have played a major role as a communicative system in the last thirty years, the internet and the web have not truly changed how peer-reviewed publishing works. Instead, many journals continue to operate largely on a print-based workflow even if we, the authors and editors who contribute to our various scholarly and scientific discussions, mostly work through email and online systems that manage the peer review and publication processes. These systems mimic the print-based workflows journals used before the internet and the web, but these systems have not been transformative even though they involve digital technologies. Authors may type papers in word processing applications, save those papers as document files (e.g., DOCX), submit those files to journal manuscript systems, and then receive decisions or reviews via those systems. However, the basic processes are similar to the basic processes used before the internet and the web. These processes closely mimic the typewriter, the copier, and the postal service; they do not radically transform the system, nor do they take advantage of what digital technologies offer.

The internet and the web have also not substantially changed how journals publish articles. Many journals, even ones that are completely digital or were born digital, continue to publish journal issues. This is despite the fact that the journal issue was a device used to bundle journal articles for hard copy printing. Printing hard copy journal articles as issues made economical sense. Bundling is a strategy to reduce cost by packaging multiple products, such as journal articles, as one product, such as journal issues. Bundling articles into journal issues also made sense given the workflow with analog technologies available since the dawn of the scientific journal up to the invention of the internet and the web. It simply made sense, in the print era, to mail issues (collections of articles) rather than to mail individual articles.

The influence of the print era on current peer-reviewed publishing practices cannot be underestimated. As Bartling & Friesike [1] wrote, the publishing culture "is affected by the journal system created when results simply had to be printed on paper," and that now, despite having digital affordances at our disposal that are limited only by our imagination, "we are currently in a 'legacy gap'".

Since we continue to operate as if we use typewriters to write papers and the postal service to manage communication about the submission and publication process, I think it is safe to claim that we still reside in the print era. Although we have gone digital, we have gone digital superficially. The technologies and workflows we use now are merely simulacra of what we used before the internet and the web [2]. I find it easy to believe that if we transported a scientist from the 1970s to today and introduced them to the processes we use in 2023, they would catch on pretty quickly because the model is plainly the same.

I believe that some of the big problems that science has today are the result of mimicking those print-based workflows within a digital environment. The open access movement is concerned with making the journal article and other traditional scholarly outputs more accessible. The open science movement goes further. It is an attempt to make the entire research process more transparent [1]. This is achieved by disseminating and communicating as much of the research workflow as possible, which was not possible at scale in the print era. Although there have always been barriers to access, the goal of publishing research has long been fundamentally communicative and about knowledge sharing. The promise with digital technologies is to provide access to more of the research process than analog technologies could in the print era. However, since journals continue to function using a print era model, they will be hard pressed to truly realize the full potential of open science.

There are many ways we can embrace digital technologies that foster our pursuit for greater transparency in science. To accomplish this goal, the processes that we use to communicate our research should be re-evaluated based on the evidence and with the goal to improve science and its dissemination. Nothing we do should be left unexamined or unquestioned. Since all technology offers affordances, where such affordances influence how we act and what we think is possible, we should reconsider the technologies that we use to review our manuscripts, communicate our research, structure our papers, disseminate those papers, organize our end products, and so on.

In this paper, though, my wish is to focus on the journal issue and the problems it presents to us as a relic of the print era. I also wish to present other ways of bundling our output that more fully embrace digital technologies. My solutions are only suggestions, and I raise them to stimulate discussion. However, there are two points I want to make. First, the current way we mimic the print era in our publication workflow is detrimental to the pursuit of not just science but also to the pursuit of open science. Second, and more broadly, instead of leaving progress in publishing up to random agents, as explored by Binfield [3], we should collectively and rationally examine our methods in order to imagine and enact improvements [4]. As a proposal, I offer for discussion the idea that journals could re-envision themselves as digital libraries instead of as periodicals.

## 2. The Journal Issue

Bundling by journal issue has had ramifications on everything from how scientific output has been organized to how it is retrieved. In this section, I discuss two overarching ways that journal issues have harmed the communication of science, and therefore, science itself. First, I discuss how the print era of bundling created real scarcity and how that scarcity unjustifiably persisted with the introduction of digital technologies. Second, I discuss how journal publishers have created complexity by publishing multiple versions of articles that attempt to satisfy print era workflows.

## 3. Scarcity

Journal articles were bundled by issue because they were modeled after early newspapers [5]. It was economical to release a set of articles in an issue, rather than mailing each article to subscribers one-by-one [6]. Printing costs limited the number of pages that journals could print in a single year [7]. This constraint created scarcity, and scarcity meant that editors had to make decisions about what to print and what not to print. Even if the intentions were good, this editorial gatekeeping due to scarcity probably contributed to the

increasing disappearance of papers that report negative results and the recognition of a reproducibility crisis in some scientific disciplines [8,9].

Scarcity creates value, and this is true for scholarly publishing [10,11]. One metric of value is the acceptance rate, which functions as a signal of a commodity's value and unavailability. The lower the acceptance rate, the higher the scarcity value. The acceptance rate of a journal has long been used as an indicator of a journal's perceived quality or its value [12]. A journal with a 10% acceptance rate is held in higher esteem, all things being equal, than a journal with a 50% acceptance rate. However, the acceptance rate of a journal is a function of the print model [13]. When space is limited (available print pages per year), i.e., when a journal has page constraints to consider, then its real estate becomes, by this constraint, prized [14]. Although page constraints do not entirely explain acceptance rates, Björk [15] showed that selectivity based on quality of research can be sustained when page constraints are not a factor in acceptance.

Journal-based metrics, such as the Journal Impact Factor (JIF), are based on this scarcity value. This means that bundling by issue has influenced the evaluation of scholarship. The JIF is a problematic measure of a journal's impact [16], but it is a measure that only, at best, makes sense when a journal publishes a fairly constant set of articles per year. That is, in order to measure changes in average citation counts, whether it is an average over a two- or five-year period, it helps to hold at a constant the rate of citable items (e.g., articles) published per year. If the rate of articles that are published each year varies considerably, then it becomes meaningless to use the JIF to compare average citation rates across years for a single journal.

Seeking higher JIF scores therefore becomes a motivation to keep the number of published articles relatively fixed at the same number each year. The only reason to keep the number of articles each year relatively static, especially with digital only journals, is because the JIF is determined by the ratio of citations to total citable items within a time range. If journals increase their publication output by eliminating journal issues and not controlling acceptance rates based on print constraints, then their JIF scores may drop. This may be why journals keep producing hard copies even if it is no longer necessary or continue to bundle articles into issues [17].

Outside of the metrics issues, controlling scarcity has had a major impact on the production of knowledge. It determines what can be printed, and it determines how many journals exist. Journals are created because of supply and demand. If one journal is too restrictive and offers less space, then it means that other, less restrictive journals are created to meet demand.

## 4. The Bibliographic Record

Bundling by journal issue has had ramifications on bibliographic data, the records used to organize information and aid information retrieval. Journals that continue to publish hard copies, and therefore, publish by issue, often publish articles as online-first or as ahead-of-print articles, and think this is satisfactory [14]. These online-first articles become attached to journal issues at some point in their future, usually when the journal has caught up with their backlog. The result is that these articles become versioned. The versioning ranges from articles that are online-first, fully formatted, and not attached to a journal issue, to articles that become attached to a journal issue. This result means that such articles receive at least two publication dates. The first date reflects the online-first version. The second date reflects the date of the journal issue. The time difference between these two versions may be years.

These dates are recorded in the bibliographic record, and consequently, this impacts information search and retrieval. Consider two of the U.S. National Library of Medicine's PubMed's date search fields: Date of Electronic Publication (DEP) and Date of Publication (DP). The DEP marks "the date the publisher made an electronic version of the article available." Therefore, the DEP can be the date an article was made available online-first and not attached to an issue. The DP "contains the full date on which the issue of the journal

was published". This means that some journal articles may have at least two publication dates. For example, suppose an article is accepted by a journal and then is quickly made available as an early access or online-first article on the journal's website. The date that it is made available is the DEP. Later, the article is assigned to a journal issue, and when that journal issue is released, the article is given the DP. The bibliographic record for such an article records these two dates. Each date may be selected in a PubMed search since these are different date fields in PubMed, but using one search field rather than the other may mean unintentionally excluding works that have already been published.

This is an example of mimicking a print-based way of doing things that is complicated by a digital-based way of doing things. By mimicking a print-based workflow in our digital settings, extra steps have been added to ill effect. In our paper on the reproducibility of search queries across different MEDLINE platforms [18], we showed that a paper on cancer made available online-first in 2015 was not assigned to a journal issue until 2019. The online-first version of the paper was available in PubMed soon after it was available on the journal's website as an online-first article, but the four-year delay to assign the paper to an issue meant that the paper was not available in PubMed's MEDLINE subset during that four-year period. The paper was therefore invisible to cancer researchers who use MEDLINE and its controlled vocabulary to finely control their bibliographic searches. (Our paper, on the other hand, was fairly quickly assigned subject headings in MEDLINE soon after publication since it was published in a journal (PLOS) that does not publish by issue in the conventional way). Furthermore, since MEDLINE is available on multiple database platforms other than PubMed, such as Ovid, Web of Science, EBSCOhost, and ProQuest, we found that for MEDLINE searches limited by dates at least four years prior to the searches, bibliographic databases constantly varied the number of search results over the course of a year of searching because of likely changes to the bibliographic records. This is surprising. In the print era, bibliographic records were largely fixed (the records were printed as hard copies). Because many journals, especially those that continue to print hard copies, try to have it both ways (print and digital workflows), the result is, as described above, an over-complicated search system. This kind of complication is strictly the result of cross-walking a print-based workflow onto a digital-based workflow.

It would be incorrect to assign fault to the National Library of Medicine's method of managing bibliographic records in PubMed and MEDLINE. Librarians' roles are to catalog, document, and organize publishing information for retrieval. When that publishing information changes, librarians respond by updating the bibliographic records. The problem is that the records should not change. If the journal for the article above ceased mimicking a print-based workflow, by publishing articles online-first and then later in journal issues, then this versioning problem would not exist. The journal could stop mimicking a print-based workflow by ceasing to version their articles based on when they become available online, as online-first articles, and then by issue and volume numbers. Instead, they can publish after peer-review and formatting and drop the production of journal issues. Issue and volume numbers are strictly a print-based artifact, put in use to bundle articles into booklets for snail-based mail. Journals that do not mimic the print-based workflow in this way do not have this problem and may avoid the illusion of scarcity that was real in the print era. Often these are journals that were created after the web started and that do not print issues (e.g., consider PeerJ.com (accessed on 31 December 2022)).

## 5. Solutions

Up to this point, I have described some problems that are caused by journals mimicking an outdated print-based workflow and how these print-based systems are harmful to science and its dissemination. In this section, I describe solutions that align with digital publishing. These solutions would benefit the dissemination of science and therefore science itself.

Literature is discovered today largely through general and special search engines, bibliographic databases, and social media [19]. Few people, it seems, discover new research

by perusing the table of contents, which again are themselves ordered by when articles are ready to publish, and not ordered thematically or topically. Although websites exist that collate articles by topic [20], there is no reason for journals to continue the practice of creating tables of contents. It is a fundamentally poor way to collect and organize unique information sources such as articles, especially for journals that are completely digital.

Instead, digital journals can create and curate exhibitions and collections similar to how the Digital Public Library of America (dp.la) presents exhibitions and collections of its partner institutions, or how PeerJ and PLOS present collections of articles. In doing so, journals can become digital libraries. This has several benefits. Digital libraries have the advantage of offering multiple ways to organize and classify the works in their collections (as opposed to a linear and time-dependent issue and volume system), and this creates opportunities to forge and design web interfaces that match those collections and that make browsing a first-rate activity [21]. For example, articles can be assigned to multiple exhibitions and collections based on their main and secondary topics. By changing to this practice, journals could increase search engine discoverability and foster browsing and perusal. Browsing and perusal have long been just as important aspects of information retrieval as query-based search has been [22,23].

In short, journal websites could function as digital libraries instead of mimicking the format and practices of hard copy journal issues and publishing. Just as librarians assign multiple subject headings, and thereby provide multiple access points, to works, journal articles could be placed in multiple exhibits and collections, thereby increasing the number of access points and the discovery of them. This becomes a matter of web interface design, information architecture, and curationrather than simple article bundling based on a first-in, first-out table of contents print-based model [21]. Even for journals that need to print hard copies, it is still possible for them to give primacy to a digital workflow rather than a print workflow. More pointedly, their websites could function as digital libraries, as described above, even if they continue to print journal issues in the traditional way.

We might conclude that many journals, as products of the print era or as adopters of print workflows, as most continue to exist today, are themselves obsolete. The future of scientific publishing could be based on creating and curating digital libraries of scientific and other scholarly output. Other improvements can be made, too, in the spirit of reflecting on our practices. Articles themselves can be re-evaluated [24]. Many journals to this day publish tables in articles not as machine-readable HTML tables but as PDF, JPG, or PNG files, which makes them largely inaccessible to data extraction and introduces possible sources of error in meta-analyses which need tabular data from multiple sources to synthesize statistical calculations. Although services such as Unpaywall help discover open access versions of articles, journals, as digital libraries, could develop better relationships with preprint archives and establish a chain of providence that links preprints to their peer-reviewed outputs. Other outputs could be more obviously connected to journal articles, such as data journals and computational notebooks [25]. Those outputs could become part of the digital library's collection and therefore part of the bibliographic record. For example, this paper was written in Markdown and synced to a repository on my GitHub account, which could be a legitimate part of a modern, digital scholarly workflow [26,27]. Acceptance rates, as a function of scarcity, could become a thing of the past since digital space, theoretically, is unlimited, as the journal eLife is exploring [28].

## 6. Conclusions

Science depends on a communication system, and the current communication system is largely based on the internet and the web. Despite that, much of scholarly publishing continues to function as if it were still in the print era. This is evident in the way journals continue to publish, for example, by bundling articles into journal issues. A host of problems arises from this that impact what journals publish, how metrics are calculated, how acceptance rates become prestige markers, how information is organized and retrieved, and how knowledge is shared.

I offered some solutions to improve the way science and scholarship is published. My goal is to foster discussion since actual solutions depend on the needs of the communities involved and not on the desiderata of any specific individual. However, I do believe that my main solution, re-conceiving the journal as a digital library, is more aligned with what the internet and the web affords, and that continuing to apply print era workflows and practices to scholarly publishing is harmful.

In the end, solutions must be rational, evidence-based, reflective, aware of scholarly workflows and interconnections, and solved collectively. The internet and the web have been disruptive, but the transition to digital publishing has been slow and left to individual entities to make progress. I believe that we should always ask ourselves some basic questions: What best serves scholarly communication and knowledge sharing? Does our current system support what is best? How does 'what is best' vary by discipline?

When digital publishing became available, the affordances offered by print could be copied over to online systems that manage manuscript submissions. This is because these online systems and digital technologies are flexible and can encompass and mimic print-based affordances. However, digital publishing can afford much more and can better serve science, scholarship, and knowledge sharing.

In summary, journals could start imagining themselves as part of the larger scholarly web, which itself was designed to be interconnected, instead of designing their sites as silos that consider the journal article as the final, definitive, machine-unreadable end product. Current scholarly publishing is woefully outdated and remains loyal to print era workflows and processes. By embracing the digital, we can avoid multiple publishing dates, give primacy to HTML output, make articles sources of data (machine readable) and not merely sources for reading, create greater interoperability, and eliminate the requirement to submit manuscripts as word processing files [29]. With these and other improvements, such a system would be truly knowledge-producing.

**Funding:** This research received no external funding.

**Institutional Review Board Statement:** Not applicable.

**Informed Consent Statement:** Not applicable.

**Data Availability Statement:** Not applicable.

**Acknowledgments:** The author would like to thank Daniela Di Giacomo for her comments on a draft that helped improve this manuscript.

**Conflicts of Interest:** The authors declare no conflict of interest.

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
