# Peer review of "The Issues with Journal Issues: Let Journals Be Digital Libraries"

_publications, doi:10.3390/publications11010007_

Round 1

Reviewer 1 Report

The topic is relevant: the role of academic journals in the system of scientific communication, and their contribution to the future development of science, especially in the context of open science. The paper is a viewpoint, not a research paper or a systematic review; this means that most usual criteria of peer reviews do not apply. 

The paper is rather easy to read, even a little bit lengthy for an opinion paper. The main argument - that the actuel way of making academic journals is detrimental of open science - is repeated a couple of times but remains rather general and vague and should be sharpened. I'll try to sum up my concerns in three points.

First, the main argument and the suggested solution - the transformation of journals into digital libraries - is anything but new or original. This suggestion has been made since the launch of the web, thirty years ago. How many times have I read comments, papers, viewpoints, analyses, predictions like: journals risk dying; journals may die; they will die; journals should die... Especially in the context of open science and the development of open repositories and preprint servers, this was and still is a kind of mantra, and this was and is also the reason and rationale for many initiatives of publishing results. My concern is that the author proposes a more or less ahistorical viewpoint without a real understanding of the development of academic journals before and after the launch of the web. This may be the reason why new models of academic publishing like open repositories, overlay journals or preprint servers are nearly completely missing. Also, I think that the author underestimates the actual transformation of journal platforms from Elsevier, Springer Nature and others into something very different from digital libraries (see for instance the progressive vertical integration of the Elsevier products or of Digital Science). Reading the viewpoint, I sometimes had the feeling that the author was kicking at an open door and preaching the converted; or worse, that he was trying to open the wrong door.

Second, the paper lacks a kind of strategic analysis of the situation of the academic journals. If it is true that journals are dysfunctional and obsolete ("detrimental"), so why are they still in the field? Why is their number continuously increasing? The only real argument for their obsolescence I can find is that a paper (the Pubmed and Medline example) may be available on one server or platform years before it is also available on others. So what? Is this really a great problem? (Also, the author mentions "some of the big problems of science" - which problems???). My impression (based on recent studies) is that journals may have many flaws and problems (and of course, they have) but that they still fulfill some significant functions in the system of scientific communication - beyond the traditional "Oldenburg functions", they produce for instance money, data and metrics, they (still) structure communities etc. So, I would suggest that the author includes some more references on journal publishing and tries to understand why the journals are still there. This would provide depth to his viewpoint that journals should be transformed. 

Third, and related to my second point, I would also suggest a kind of SWOT analysis which would be helpful for the description and understanding of dynamics and barriers in the field of journal publishing. In fact, my impression is that the main problem may not be the bundling (issues) but the (lack of) integration into a system of papers, data and other materials, research assessment and so on, a problem in terms of standards, formats, interoperability, metadata, identifiers etc. 

More generally, don't let me be misunderstood: I think that the viewpoint is quite legitimate but I think, too, that the arguments are weak and that the paper does not contribute to the debate on academic publishing with new elements. Another, general issue is that some sentences (especially in the first part) are assumptions without any evidence, and that some terms lacks precision (eg., true digital libraries - what are untrue/false digital libraries???).  

Reviewer 2 Report

I undertand this is not a research article but a viewpoint, therefore the review has to be performed accordingly

In general I think it is well written and I agree with many of the arguments explained there.

Just a few comments, I would tend not to use specific file formats when mentioning document files (row 28). I rather use a general term like "document files" knowing that in many journals formats used are diverse.

I guess some readers will argue that another reason why there is a low level of acceptance is the need to have enough resources to manage a high level of articles. I don't know if you could comment on that, too.

Finally, at the end of the text, just before the conclusion there is a paragraph commenting on other issues where journals need a huge improvement. It is a bit telegraphic and there is only a list of elements to improve but maybe there is a lack of possible solutions. For instance, which is the solution for publishing tables? Change format? Link datasets?

I hope you could address these comments. Thanks

Reviewer 3 Report

The article deals with the topic effectively, the research question is relevant and interesting.

The argument is original, not widespread in the specialized literature, though covered in several online posts. The analysis is capable of capturing relevant aspects for scholarly communication. The proposed solutions are clear and well argued.

The article deals with the topic effectively, the research question is relevant and interesting.

The argument is original, not widespread in the specialized literature, though covered in several online posts. The analysis can capture relevant aspects of current scholarly communication. The discussion is quite well argued.

In support of the thesis of the paper, some topics or references can be inserted in a concise way, without changing the structure of the discussion.

(26) (95) One or more references regarding the historical formation of the issue-based system in academic publishing may be required to properly analyze and discuss about its development. The identification of the principle of scarcity as a lever for maintaining a model is relevant in the discussion. But it should perhaps be contextualized in the use of a pre-existing model.

(i.e. 201) Furthermore, it would be necessary to refer to journals that have already adopted a different publication model (if they have a JIF it would be useful to briefly analyze its trend), for example Plos One, PeerJ journals (which are cited by the author regarding indexing issues in MEDLINE).

(195) The author focuses on cons of information retrieval and some aspects of metadating and indexing: in this regard it may be useful to mention the DOI and the article number, which have become mandatory-like metadata fields for journals that have gone beyond the system based on issues. PIDs and IDs are relevant in analyzing any topic related to bibliographic citation and information retrieval. It may be useful to add some information on journals that are indexed in MEDLINE while not having a issue-based publishing.

(238) Nowadays, some online services, ie. like Unpaywall, are adding value to paper versioning. Unpaywall is linking available open versions to articles, both natively on some publisher sites, or by browser extension, and its information are displayed databases such as Scopus, Web of Science, EPMC. Many tools or discovery services are already integrating this feature.

(245) The conclusions section does not adequately address the large number of journals that already make full-text articles available in HTML, both on proprietary platforms or through OJS. Obviously, the availability of full-text, and its contents, depends much more on accessibility (paywall or open access) than on the formats in which they are released. The conclusions both on machine readable availability and on the submission and reading systems may seem a bit dated compared to the publication environments used by many readers and authors. Perhaps an examination of the positive aspects (if any) for readers and authors of issue-based publishing could make the discussion more balanced. The lack of pros does not make the cons less relevant.

The related bibliography should be promptly inserted.        

The conclusions do not consider the large number of journals that already make full-text articles available in HTML, both on proprietary platforms and through OJS. The conclusions both on machine readable availability and on the submission and reading systems may seem a bit dated compared to the publication environments used by many readers and authors. Perhaps an examination of the positive aspects (if any) of issue-based publishing could make the discussion more balanced, but the lack of pros does not make the cons less relevant.

Another topic to be treated briefly, and not only regarding e-pub ahead of print, is digital paper versioning or / and the availability of preprint servers and repository. A reference, not only in the introduction, to open science and the open access workflow would make the argument (on the cons of issue-based publishing) stronger. The relevant bibliography should be promptly inserted.
